# Modelling and improving approach for carrying capacity in multi-modal super network by considering travel time

Xiangyue Huang[ID]*

College of Transportation Engineering, Chang''an University, Xi'an, Shaanxi Province, China

* hxyue019@163.com

## Abstract

For the sake of achieving the mensuration of network carrying capacity under regular even congested road conditions with crowded vehicles, passengers or cyclists in a median-scale network, this study examined the ideal travel time of a passenger or cyclist in the hybrid-congested roads (congested and non-congested road accounts for half), being the initial accumulated values for the independent variable in Aggregated Functional Equation. Besides, the various road section's capacity was examined, together with the total travel time accumulated accordingly, and the capacity limitation took effect to alleviate the heavy load of vehicles on partial road sections. This study proposes a multi-objective bi-level planning model on the basis of the original capacity constraint model to address the potential congested problem, the model optimizes three aspects in the schematic diagram: carrying capacity, average travel time, and expansion cost. This experiment attains 9 effective hyper-paths with traffic flow and travel impedance attributes attached to each hyper-path. The scatter chart results show that the plural modulus of carrying capacity decreases with travel time's descends; when the road saturation increases from 0.7 to 0.9, the plural modulus of carrying capacity declines by 27.1%. Meanwhile, raising the upper limit of expansion would slightly lift the carrying capacity in the proportion of 15.7%. This research could provide a reference for the majorization of multi-modal cyber flow distribution and also has indirect significance for route planning to a certain extent.

## 1. Introduction

In a bid to understand and formulate the operation strategy of a multi-modal transportation system. Traffic designers were devoted to figuring out complex rules to predict congestion on the road sections [1]. The analysis of cyber arcs is beneficial to reduce drivers' travelling time and explain abstract concepts. As we all know, the single-layered network is closely linked to the corresponding transport mode. A special path searching method of a complicated network takes pains to avoid taking

**Data availability statement:** All relevant data are within the paper and its Supporting information files derived form calculation by author.

**Funding:** The author(s) received no specific funding for this work.

**Competing interests:** The author declares that there is no known competing financial interests or personal relationships that could have appeared to influence the work reported in this research article.

a detour of both one-way road networks and two-way road networks [2]. In acquiring station characteristic data, the algorithmic computation technologies shed new light on encouraging a broader transport choice [3]. Due to that, travelling impedance is one of the most significant indicators resulting in partial blockage between different modes. In addition, the dynamic changes of road traffic flow are excluded under objective reasons such as bad weather or large-scale activities, which alleviates the random fluctuation of impedance [4].

According to the average capacity constraint of each road section, optimization methods of super network design are usually composed of two steps: first step is to add up the aggregate maximum carrying capacity no matter how traffic demand varies [5]; the second step is to minimize the average travel time regarding the whole system. In this study, a bi-level planning model based on balanced traffic flow is devised with the entire carrying capacity, which is denoted as the optimal target [6]. From the perspective of theoretical research, another study has used a similar methodology to calculate the goal value [7]. Zhao [8] applied the bi-level planning model in a high-speed railway operation system for the time-varying demand. Bahk [9] and Yang [10] explored the characteristic attributes of the lane with respect to the multimodal network, and a time-space consumption method was proposed under various conditions.

When a single vehicle is running on the road, due to the limitations of the surrounding driving environment, the traffic flow scale would be affected by the road capacity. When the network scale is wide, it may cause congestion on certain road sections [11,12]. Based on the above fact, this essay carries on the reasonable lane combination scheme within the multi-modal network. The contribution of this essay is mainly generalized in four aspects:

(1) Considered the conservation of network traffic to establish a multi-modal network distribution model based on Method of Successive Algorithm, mainly to troubleshoot congested road sections and promote the balanced distribution of traffic.

(2) Established a bi-level planning model for integrated urban transportation modes, aiming to minimize travel time and maximize network carrying capacity, thus reducing traffic detours and improving driving efficiency.

(3) Obtained the flow distribution results by iterative method and incorporated them into the initial population of improved genetic algorithm, retained a method of screening out the best individuals for reproduction and keeping them to the next generation, designed a high-performance NSGA-II algorithm.

(4) Studied the relationship between road widening and cost consumption, added one of the goals to minimize investment cost, and proposed an improved multi-objective urban integrated transportation network optimization model.

This study provided the experience for the design of road improvement schemes for continuous-based network in the given case study and could serve as a good reference.

## 2. Literature review

### 2.1. Traffic assignment solvers

As people become more conscious about smart mobility, some scholars have started to analyze the effect of synthetical transport means while practicing in a two-way traffic network. Si et al. [13] constructed a mathematical model to describe the equilibrium state of Wardrop flow, laying the foundation for the construction of the multi-modal flow distribution model. The equilibrium distribution flow model is put forward to simplify the calculation, with a streamlined diagonalization algorithm. Then, the hybrid traffic super network is simulated with numerical examples, and the convergence results are verified.

Bovy [14] established a simple Frank Wolfe algorithm that can avoid the elimination of all traffic network paths to analyze the multi-modal path flow. Subsequently, Wong [15] constructed a mixed user equilibrium distribution model under combined traffic modes. By transforming the super network into the combined network including private cars, scholars proposed the mixed equilibrium prerequisite for travelers, and calculated the analytical model utilizing the method of successive algorithms. In addition, analyzing the simple influences of bike-sharing on balanced flow and total travel impedance in virtue of three modes, we finally found the optimal road widening scheme that minimized the total travel cost of the system through the algorithm's calculation [16]. Apart from that, the congestion impedance function could reveal the driving time and flow of road sections under high-delay condition. Yang [17] carried out the relevant balanced traffic distribution in the traffic network of travelers in the compeletely crowded network. Shi [18] proved that non-motorized transportation modes on passengers' journey would also take effect in a compelicated network. By dint of GPS trajectory data and POI, Lan et al. [19] constructed a dynamic traffic flow allocation model. Meanwhile, Gizem [20] put forward an optimal traffic distribution model with minimum travel time, a dynamic traffic distribution model based on simulation, and a model describing the dynamic traffic distribution of a large-scale network in view of the holographic principle of travel time (dynamic traffic assignment, DTA) [21,22].

### 2.2. Carrying capacity models

Baublys [23] established an original model of traffic carrying capacity aiming at maximizing carrying capacity, considering the selection behavior of travelers, and simultaneously catered to the constraint of road sections' capacity. Model construction should usually predict how many additional traffic demands could be fulfilled, thus tinkering with the corresponding land use development plan and road pricing scheme. By the aid of this pattern, Brian et al. [24] analyzed the theory of synergy and the principle of composite system, mastered the tendency of the coordinated development of land use and urban system. Zhang et al. [25] utilized the coupling coordination degree measurement model of urbanization and traffic, drew support from the coupling degree function, and created the evaluation specification to illustrate the scientific research solution from numerical case valiadation.

Combined with massive coalescent theory to delve into, from the perspective of continuous-based network design, the existing exploration focuses on the carrying capacity of quantified multi-modal networks is not sufficient; it needs including further improvements to critical aptotic sections in the extant network, instead of optimizing the entire network by adding new routes. Super networks might customarily turn into cyber structures attached with associated compositions, which could be represented as small, medium or large-scale network structures abstracted from continuous lines and independent or cross-over points. In particular, it is found that the network topology configuration should significantly have an effect on the accumulation of cyber carrying capacity before the investigated data is acquired [26].

### 2.3. Integration of bi-level model and carrying capacity

Under majority of conditions, shared bikes riding are on the outermost side of roads, which lack the separation belt between non-motorized and motorized lanes, and could be apt to trigger friction among cyclists and drivers. Buses, due

---

to their large size and lofty passenger capacity, ought to require sufficient clearance at stops for passengers to get on and off. To avoid potential risks, policymakers would rather take measures such as widening lanes to keep vehicles safe. The standard bi-level planning model is composed of the upper-level model section and the lower-level model section.

As illustrated in Fig 1, experts are still dedicated to improve network carrying capacity [27,28]. The designed model regards the path impedance update as the lower-level model's target. Boarding and alighting time, average ridership, and mode selection probability in the mechanics diagram are also considered. In essay, lanes' broadening scheme is set in the upper-level part. The upper-level part is regarded as a mathematical convex optimization problem containing two or three objectives with capacity constraints. If it is assumed that the road section could accommodate infinitely large traffic flow to allow all traffic demands to access, while ensuring traveling accessibility, an Augmented Lagrangian Function would be constructed to characterize the relationship between feasible traffic flow and capacity on sections, and an external penalty factor is incorporated into the extended objective function [29]. When the objective of travel time approaches the best solution, there must generate a corresponding feasible, optimized scheme which could be adopted. In the lower-level model, adopting the travel time of users between origin and destination as an impedance for network optimization could achieve the optimal state of network traffic distribution. However, when it comes to uncertainty factors such as road saturation and expansion cost, which make it hard to change the overall traveling status in the original models.

For the bi-level planning model covering multiple variables, it is difficult to obtain an exact solution by mathematical methods such as the Lagrange method and gradient descent. Therefore, Zhang et al. [30] put forward a majorization approach of an advanced generic algorithm, called NSGA-II (non-dominating sorting genetic algorithm), to solve the constrained multi-objective network design problem and express the optimal solution of the model with a Pareto frontier relationship.

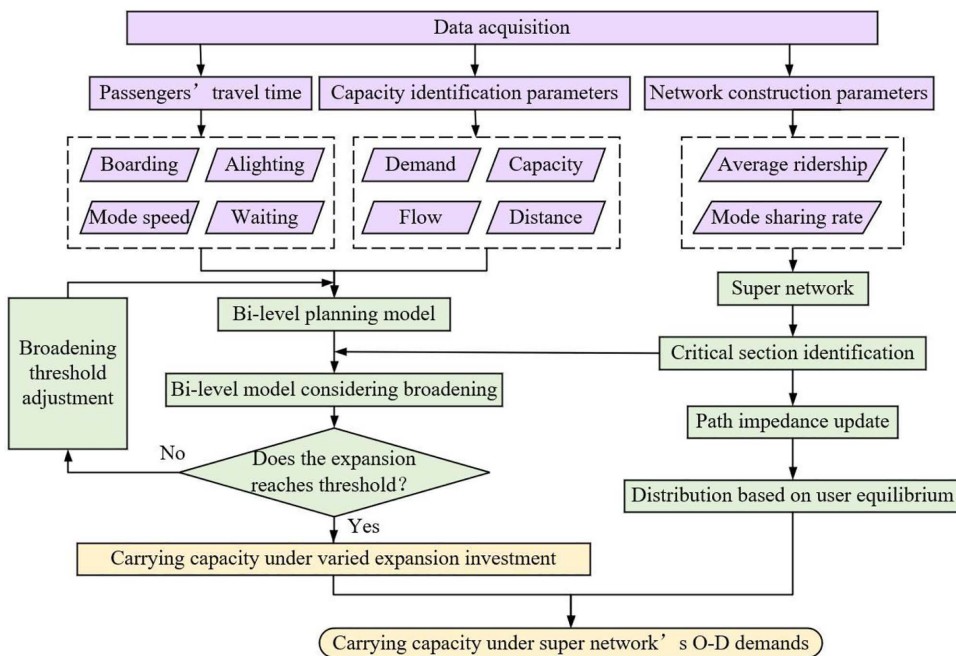

**Fig 1. Mechanism of bi-level planning model in essay.**

# 3. Methodology

## 3.1. Research scenario

The directed multi-modal transportation network $G(V, A)$ includes the basic arcs and characteristic parameters of $m$ layers' sub-networks reshaping the urban super network. Define traffic modes as $G\{G_1, G_2,..., G_{m-1}, G_m\}$, where each mode sub-network consists of station connection arcs and traveling arcs, as shown in Fig 2.

$V$ and $A$ represent the sets of all nodes and segments, respectively. Origin and destination named $r$ and $s$ separately carry the initial demand $q_{rs}$. The basic assumptions of the model are set as follows:

(1) The actual public transport system may operate later than the scheduled timetable;

(2) This research would not absorb certain physical parameters in the paper, such as transit time, driving speed, and vehicle weights; only the waiting and walking time on the station connection arcs, traveling arcs are included, indicating that any transferring time equals zero [31,32];

(3) In one journey, among all possible travel paths between the origin and destination, the path that minimizes travel cost or impedance is selected by default;

(4) In the expansion plan, the number of expansions refers to the widening of different numbered sections or the repeated widening of the same numbered section.

## 3.2. Notation

To interpret the meaning of the parameters and variables in the model, descriptions of parameters and variable symbols are given in Tables 1 and 2.

## 3.3. The upper-model analysis

The upper-level counterpart of the bi-level planning model has its own objectives and constraints.

**3.3.1. Objective function.** This section calculates the weighted sum of the carrying capacity among three separate networks from both sides of the diagram, aiming at widening critical road sections. The multi-objective function could be divided into multiple counterparts, that is, $F(x)=\omega_1 f_1(x)+...+\omega_i f_i(x)$, where $\omega_1,..., \omega_i$ equal 1 at the beginning, and the final values indicate the weights of their corresponding function. When the value of $i$ equals 2, the original function converts

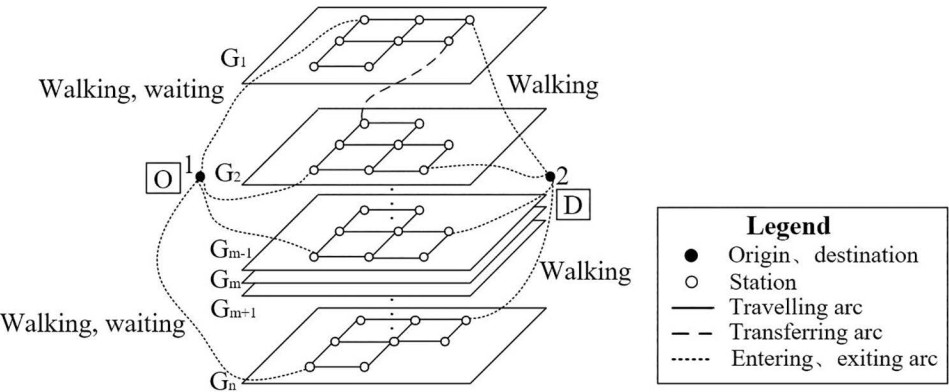

**Fig 2. A schematic diagram of the case network topology.**

**Table 1. Interpretation of model's main parameters.**

| Parameters | Unit | Parameters' definition |
|---|---|---|
| $V$ | – | Set of nodes in the multi-modal network |
| $A$ | – | Set of road sections in the multi-modal network |
| $K$ | – | Set of paths in the multi-modal network |
| $M$ | – | Set of modes in the multi-modal network |
| $I$ | – | Set of upper and lower arcs |
| $G_m$ | – | Single-mode in the sub-network |
| $O$ | – | Origin in the multi-modal network |
| $D$ | – | Destination in the multi-modal network |
| $k_i$ | % | Sharing ratio attached to random layer's network |
| $l_m$ | m | Length of the expanded road section |
| $l_a$ | m | Length of travelling arc in multi-modal network |
| $v_a$ | m/s | Walking speed of the travellers on entering and exiting arcs |
| $v_m$ | pcu | Standard vehicles' conversion coefficient |
| $\varepsilon$ | pcu/h | The unit investment ratio coefficient |
| $Z_m$ | ppl | Average ridership of each mode |
| $A_{m,l}$ | – | The scheduled leaving time of mode $m$ |
| $B_{m,l}$ | – | The scheduled arrival time of mode $m$ |
| $k_c$ | pcu/h | Maximum capacity of road section |
| $k_b$ | pcu/h | Widening upper limits |
| $k_m$ | 10K | Widening upper cost limits |
| $C_{m,a}$ | pcu/h | Initial capacity on section $a$, mode $m$ |

**Table 2. Interpretation of model's main variables.**

| Variables | Unit | Variables' definition |
|---|---|---|
| $T_{m,a}$ | min | Travelling impedance on section $a$, mode $m$ |
| $t_{m,0}$ | min | Free-flow travel time on section $a$, mode $m$ |
| $t_{m,1}$ | min | Upper and lower arcs' walking time |
| $t_{m,2}$ | min | Waiting time on the upper arc |
| $Z$ | min | Total impedance of passenger's travel process |
| $Q$ | pcu/h | Carrying capacity in a multi-modal network |
| $\bar{t}$ | min | Individual's average travel time |
| $\alpha$ | – | Broadening quantity of capacity |
| $C$ | pcu/h | Capacity after expansion on section $a$, mode $m$ |
| $\sigma$ | % | Network's average saturation |
| $x_{m,a}$ | pcu/h | Flow on section $a$, mode $m$ |
| $f_{m,k}$ | pcu/h | Searched path flow on mode $m$ |
| $q_{rs,m}$ | pcu/h | Travel demand of mode m between origin and destination |
| $q_{rs}$ | pcu/h | Total travel demand between origin and destination |
| $\delta_{rs,a}^k$ | – | Road section whether to go through the path, equals 1 or 0 |
| $U_{m,a}$ | min | Travelling time for entering the multi-modal network |
| $L_{m,a}$ | min | Travelling time for exiting the multi-modal network |
| $n$ | 10K/(m/ln) | Number of times for certain expanded sections |
| $y_a$ | – | Optimization scheme concerning expansion |
| $g_a$ | 10K | Investment cost concerning expansion |

to a bi-objective problem: $F(x)=\omega_1 f_1(x)+...+\omega_2 f_2(x)$; when the value of $i$ equals 3, the original function converts to a triple-objective problem: $F(x)=\omega_1 f_1(x)+\omega_2 f_2(x)+\omega_3 f_3(x)$. In the polynomial formula, $f_1(x)$ indicates the first function of the total traffic demand, its value equals $q_{rs}$ between origin and destination, or substitute indicator $Q$, the objective function is expressed as follows:

$$max f_1(x) = \sum_r \sum_s q_{rs}$$

(1)

$f_2(x)$ indicates the second function referring to average travel time measured by every transportation mode, and $k_i$ is introduced to decide the significant degree of each transport mode, and its unit is %. The objective function is described as follows:

$$min f_2(x) = \bar{t}$$

(2)

$$\bar{t} = k_1 T_{taxi,\,a} + k_2 T_{bus,\,a} + k_3 T_{bike-sharing,\,a}$$

(3)

In the formula, $f_2(x)$ represents the total travel time in three cases, with units of min. $k_1$, $k_2$ and $k_3$ respectively represent the sharing rate of taxis, shared bikes, and the buses, with units of %.

$f_3(x)$ indicates the third function of the expansion cost. Expansion cost is introduced to reflect the transportation effect to lift the carrying capacity. The third function is composed of 3 variables, that is, $n$, $\varepsilon$, and $I_m$. Where $n$ represents the number of road widening times, unit is 10K/(m/ln), $\varepsilon$ represents the unit investment ratio coefficient, and its unit is pcu/h, $I_m$ represents the widening mileage length of mode $m$, unit is $m$. The objective function is expressed as follows:

$$min f_3(x) = n\varepsilon \sum_{m \in M} I_m$$

(4)

**3.3.2. Constraints.** With the assumption that drivers choose their own travel mode without any preferences, the possibility $P_{rs}(y=m)$ in discrete selection model could be expressed as:

$$P_{r,s}(y = m) = \frac{exp(V_{mi})}{\sum_{m \in M} exp(V_{mi})}$$

(5)

$$\sum_{m \in M} P_{r,s}(y = m) = 1$$

(6)

Similarly to the formation of the distribution of the discrete selection model, the average cyber service level $\sigma$ could be expressed by the following Equation (7), explaining Equation (9) for eliminating the peak hour's jam; the common formulas that account for the probable lane width are as follows:

$$\sigma = \frac{x_{m,a}}{C_{m,a}} \times 100\%, 0 < \sigma \leq 1$$

(7)

$$C \leq C_{m,a} + \alpha$$

(8)

$$\begin{cases} C_{m,a} \leq k_c \\ 0 \leq \alpha < k_b \end{cases}$$

(9)

To ensure that the users' waiting timespan longer than the timespan between the previous and the subsequent vehicle, together with the following time length constraints attached to different arcs are as follows:

$$t_{m2,a} \geq B_{m,l} - A_{m,l} \tag{10}$$

$$\begin{cases} 0 \leq t_{m1,a} < \frac{l_{m,a}}{v_a} \\ 0 \leq t_{m2,a}, m \in M \end{cases} \tag{11}$$

Where mode $m$ includes the means of bus and taxi. Constraint (12) limits the total expansion investment within the project budget, because only one widening scheme is optimal imposed on the third function formula:

$$\sum_{a \in A} g_a(y_a) \leq k_m \tag{12}$$

Constraint (13) restricts the value to positive numbers; constraint (14) restricts the value to floating points; constraint (15) indicates that the binary variables take the value of 0 or 1.

$$x_{m,a}, f_{m,k}, C, l_m, k_a, k_b, k_m, t_{m1,a}, t_{m2,a} > 0 \tag{13}$$

$$q_{rs}, \bar{t}, g_a(y_a), \quad floating \ point \tag{14}$$

$$\delta_{rs,a}^k \quad 0/1 \tag{15}$$

### 3.4. The lower-model analysis

$$L_{m,a} = t_{m1,a} \tag{16}$$

$$t_{m1,a} = \frac{l_a}{v_a} \tag{17}$$

$$U_{m,a} = t_{m1,a} + t_{m2,a} \tag{18}$$

The boarding time and alighting time on the station connection arcs are introduced to guarantee accessibility to any sub-network. The boarding time is classified into two parts, among which, $t_{m1,a}$ represents the online walking time required to take mode $m$ from the starting point; $t_{m2,a}$ represents the average waiting time required to wait for mode m at the station; $l_a$ represents the length of section $a$, and its unit is m; $v_a$ represents the walking speed of passengers, and its unit is m/s.

Similar to the BPR (Bureau pf Public Roads) function's expression, the total travel impedance is described as follows:

$$T_{m,a} = t_{m0,a} \left[ 1 + \alpha \left( \frac{x_{m,a} \nu^m}{Z_m C_{m,a}} \right)^\beta \right] \tag{19}$$

Where $v^m$ indicates the conversion factor of the standard traffic volume; $Z_m$ indicates the average number of passengers in the traffic mode $m$; $T_{m,a}$ manages through the entire journey cost spent on the traveling arcs. To Simplify Equation (19) to highlight average road saturation, the majorizing function is as follows:

$$T_{m,a} = t_{m0,a}\left[1 + \alpha(\frac{v^m}{Z_m} \cdot \sigma)^\beta\right]$$

(20)

Where $\alpha$ and $\beta$ represent the undetermined coefficients, which equal 0.15 and 4. $v_{taxi}$, $v_{bus}$, $v_{bike-sharing}$ are set as 1, 1.75, and 0.25. $Z_{bus}$, $Z_{taxi}$, and $Z_{bike-sharing}$ equals 30, 4 and 1, respectively.

Z(f) is utilized to describe the above-mentioned target of impedance. Therefore, the impedance function is divided into three components, that is, $U_{m,a}$, $L_{m,a}$, and $T_{m,a}$. The general formula for the accurate cost of a super network is:

$$minZ(f) = \sum_{a \in A} \int_0^{x_a} t_a(\omega)dx$$

(21)

$$Z = U_{m,a} + L_{m,a} + T_{m,a}$$

(22)

$$c_{rs}^k = \sum_{a \in A} Z \cdot \delta_{rs,a}^k$$

(23)

According to the principle of flow conservation, the traffic stream that flows through the cyber section, path, and origin-destination could be described by Equations (24)–(26) as follows:

$$\sum_{m \in M} q_{rs,m} = q_{rs}$$

(24)

$$\sum_r \sum_s f_{m,k} = q_{rs,m}$$

(25)

$$x_{m,a} = \sum_r \sum_s f_{m,k}\delta_{rs,a}^k$$

(26)

Constraint (27) restricts the value to positive numbers; constraint (28) restricts the value to floating point; constraint (29) indicates that the binary variable should take the value of 0 or 1.

$$U_{m,a}, L_{m,a}, T_{m,a}, q_{rs}, f_{m,k}, x_{m,a} > 0$$

(27)

$$T_{m,a}, \quad floating\ point$$

(28)

$$\delta_{rs,a}^k \quad 0/1$$

(29)

## 4. Algorithm design

This section solves the carrying capacity calculation problem of the bi-level planning model by establishing the continuous average iteration and the non-dominant sorting genetic joint algorithm, and presents the solution of the optimization scheme with Pareto solution set.

### 4.1. Continuous averaging algorithm

Continuous averaging algorithm (Method of Successive Algorithm, MSA) is a method for figuring out multi-objective and multi-constraint nonlinear planning problems. By averaging a series of traffic or impedances obtained in the multi-modal traffic network, among which these auxiliary points are obtained by solving the auxiliary planning problem. While solving the traffic distribution, the problem is converted into a linear planning problem, therefore, this essay improves the MSA algorithm and proposes the following improved distribution algorithm. The specific steps are as follows:

Step 1: Initialization. According to the initial cost of each arc segment and sub-path, the effective hyper-path search algorithm is used to find the set of effective hyper paths in the multi-modal network, and then a random distribution of the effective hyper paths is carried out to get the distributed traffic volume of each arc segment $x_{m1,a}$, and the number of iterations n = 1 is set;

Step 2: According to the current traffic volume of each arc segment $x_{m(n), a}$, calculate the generalized cost of each arc segment and sub-path;

Step 3: According to the generalized cost of each arc segment and sub-path, find all valid paths according to the effective path search algorithm, and randomly distribute the effective super path according to the OD demand $q_{rs}$, so as to obtain the additional traffic volume $y_{m(n), a}$ of each arc segment and sub-path;

Step 4: Iteratively update the flow rate of each arc segment according to Equation (30) and calculate:

$$x_{m(n+1), a} = x_{m(n), a} + \frac{1}{n}(y_{m(n), a} - x_{m(n), a}) \tag{30}$$

Step 5: Check the convergence of the algorithm according to Equation (31), if it satisfies:

$$\sqrt{\sum_a (x_{m(n+1), a} - x_{m(n), a})^2} / \sum_a x_{m(n), a} \le gap \tag{31}$$

Where gap measured is to reflect the degree of convergence of the algorithmic iteration. Then the algorithm ends and the distribution results of each arc segment and sub-path are obtained; otherwise n = n + 1, let and turn to Step 2.

After several iterations, the iteration ends when the decision variable x is less than the setting error. The probabilistic selection model in the combined Equation (5) finds that travelers have a higher probability of choosing a path with a lower travel impedance or a lower cost. After the iterative calculation, the traffic volume and the impedance solution distributed by each path are more balanced.

### 4.2. The Non-dominated Sorting Genetic Algorithm II

The Non-dominated Sorting Genetic Algorithm II, in order to exhibit its high-performance calculating capability, the maximum number of algorithmic iterations is set as 40, the cross probability and cross index are denoted as 0.9 and 20, and the variation index is 20. Multiple iterations shows that when the difference between two adjacent iterations is less than 0.01, and the road widening value range is [200,400], indicating that the number of population in the algorithm tends to be stable.

 

As an evolutionary algorithm to analyze multi-objective optimization problems, the core of multi-objective genetic algorithm is to coordinate the correlation among various objective functions and find the optimal solution that maximizes one particular objective function. The specific steps are shown as follows:

Step 1: Selection of the applicability function; $f_{m,k}$ and $t_a$ of the corresponding $q_{rs}$ are obtained; then the upper-level model is to calculate individual fitness and the degree of each individual whoever exceeds the constraint (if the individual is a feasible solution, the degree outside the constraint is 0). Meanwhile, when the maximum number of generations is reached, the chromosome of the highest fitness is the optimal solution to the study problem, otherwise selecting the next generation population $P_{c+1}$, which continues with the bet selection operator and the elite strategy.

Step 2: Non-dominant fast sorting; its function is to retain the good offspring in the population, the optimization algorithm reduces the complexity of the problem solving; by comparing the dominant and non-dominant associations between the individual $x_i (i=1)$ and $x_j$, if no individual $x_j$ is better than $x_i$, $x_i$ is identified as a non-dominant individual; make $i=i+1$, all non-dominant individuals are searched by comparison.

Step 3: Crowding distance ranking; After completing the non-dominated sorting, we select the top-performing individuals from $F_1$ and $F_2$ to form the next-generation Parent population $P_{(c+1)}$. To maintain numerical parity with the original Parent population $P_c$, it becomes evident that all $F_3$ members cannot be accommodated within $P_{(c+1)}$. Subsequently, we perform a crowded distance ranking on the Child generation of $F_3$. This process aims to eliminate local maximum and minimum values.

Step 4: The implementation of elite strategy (Fig 3) and the determination of crowding degree; the purpose of the "elitist preservation strategy" is to prevent the loss of the optimal individual of the current group in the next generation, which would lead to the genetic algorithm not converging to the global optimal solution. The core scheme is to directly copy the excellent individuals of the current generation into the next generation without pairing and crossing. In other words, the optimal individuals of the previous generation have taken the outstanding genes, which are worth preserving.

Step 5: Crowding degree's estimation; the resulting set indicates the first level non-dominant layer of the population, which ignores the identified individual; repeating this step to the second level non-dominant layer until the entire population is stratified as shown in Fig 4.

## 5. Case study

### 5.1. Test design

Three public transportation means in the metropolitan area are selected for setting up the sub-nets in this essay. According to field investigation and topological diagram design, the connection arcs' distances of three separate sub-networks

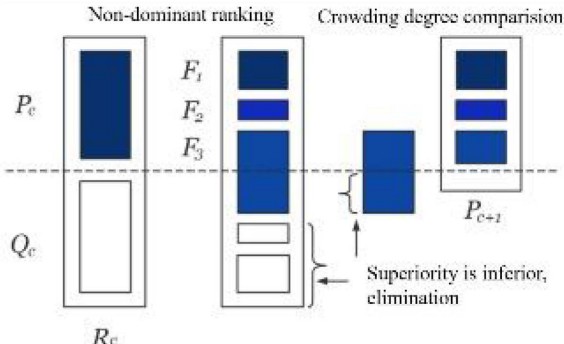

**Fig 3. The algorithm of elite strategy.**

 

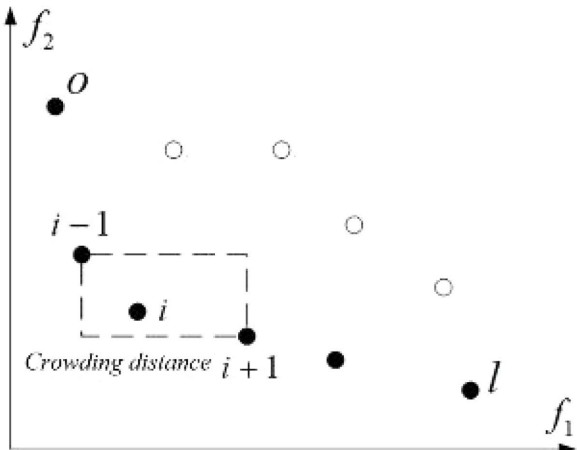

**Fig 4. The ranking based on crowding degree.**

from top to bottom are 750m, 600m, and 750m, respectively. The stations no. in the taxi network from small to large are denoted as 14, 15, 16, 17, 18, and 19 for the simplification of description. Similarly, the bus network station No. is denoted 8, 9, 10, 11, 12, and 13, and the shared bicycle network station No. is denoted 2, 3, 4, 5, 6, and 7. The topology of split sub-network and arc connection types is shown in Fig 5.

## 5.2. Data collection

Since urban roads are in general bidirectional, the transformed topology is capable of being utilized for flow analysis, congestion testing, and construction supervision. The transformation network which joined by reverse arcs based on previous indicators' setting are shown in Fig 6.

The management system has greatly improved the efficiency of transport by gathering cyber-infrastructure issues. Each mode controlls the same road combination topology, any one of which owns fixed attributes that belongs to every

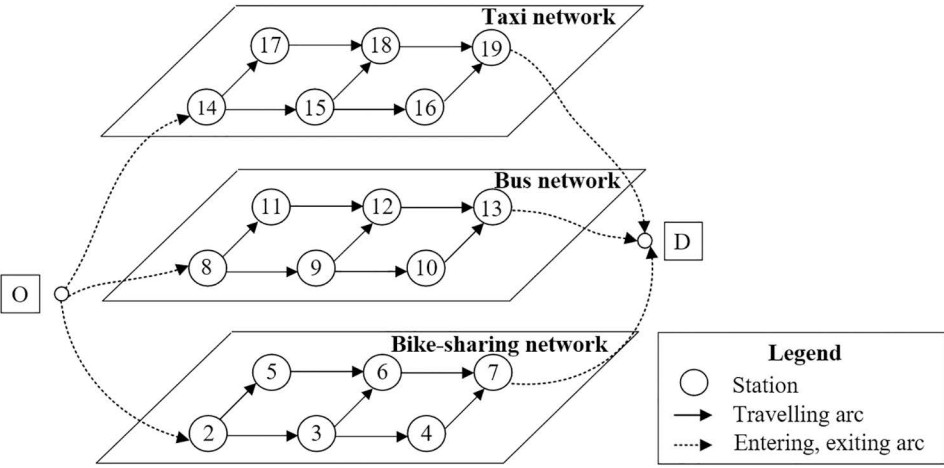

**Fig 5. Diagram of multi-modal one-way network in metropolitan.**

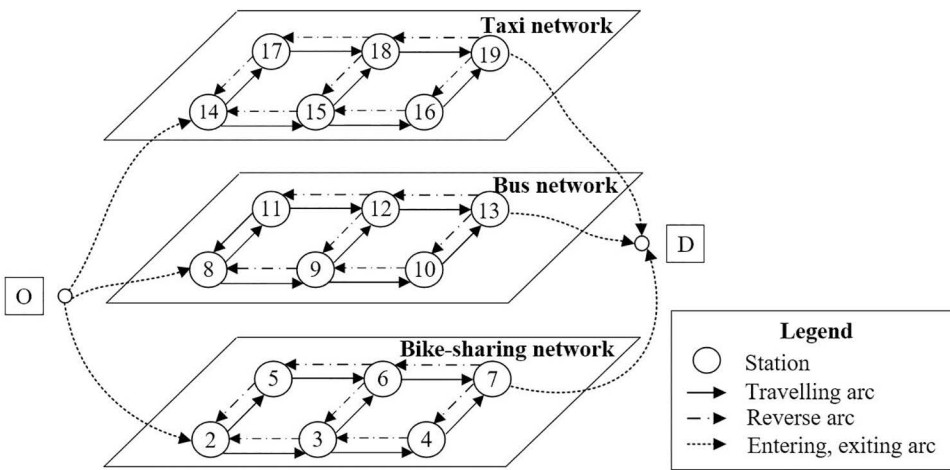

**Fig 6. Diagram of multi-modal bi-directional network in metropolitan.**

road section, such as the users' average travel time, free-flow travel time, and road section's distance attributes, partly of them are irrelevant variables to model calculations.

Taking the two-way point-to-point of home-work and home-school travel as the basis, the cost of a single car commute spends at least 15 minutes. Based on that, many scholars [33,34] have put forward the concept of time value in order to guarantee the normal operation of the city. Detailed information on the journey is given in Tables 3–6.

## 5.3. Results

Free-flow travel time, road section capacity, optimized section length, and other indicators collected were applied into continuous averaging algorithm to acquire the effective searched paths from the perspective of a construction project.

**5.3.1. Effective paths searched.** The optimal 9 paths searched between O-D pair stations are shown in Table 7.

Table 3. Travel information of taxi.

| Directed arc segments | $t_{m,0}$ (min) | $C_{m,a}$ (pcu/h) | $l_a$ (m) |
|---|---|---|---|
| 14-15 | 7.8 | 1500 | 300 |
| 14-17 | 6.0 | 1500 | 200 |
| 15-16 | 4.2 | 1800 | 300 |
| 15-18 | 3.0 | 1500 | 200 |
| 16-19 | 9.0 | 1500 | 200 |
| 17-18 | 4.8 | 1500 | 300 |
| 18-19 | 7.2 | 1500 | 300 |
| 15−14 | 7.8 | 1500 | 300 |
| 17−14 | 6.0 | 1500 | 200 |
| 16−15 | 4.2 | 1800 | 300 |
| 18−15 | 3.0 | 1500 | 200 |
| 18−17 | 4.8 | 1500 | 300 |
| 19−16 | 9.0 | 1500 | 200 |
| 19−18 | 7.2 | 1500 | 300 |

**Table 4. Travel information of bus.**

| Directed arc segments | $t_{m,0}$ (min) | $C_{m,a}$ (pcu/h) | $l_a$ (m) |
|---|---|---|---|
| 8-9 | 7.8 | 1500 | 300 |
| 8-11 | 6.0 | 1800 | 200 |
| 9-10 | 4.8 | 1500 | 300 |
| 9-12 | 3.0 | 1500 | 200 |
| 10-13 | 9.0 | 1800 | 200 |
| 11-12 | 4.8 | 1500 | 300 |
| 12-13 | 6.0 | 1800 | 300 |
| 9−8 | 7.2 | 1500 | 300 |
| 11−8 | 6.0 | 1800 | 200 |
| 10−9 | 6.0 | 1500 | 300 |
| 12−9 | 3.0 | 1500 | 200 |
| 12−11 | 4.8 | 1500 | 300 |
| 13−10 | 4.8 | 1800 | 200 |
| 13−12 | 4.8 | 1500 | 300 |

**Table 5. Travel information of shared bike.**

| Directed arc segments | $t_{m,0}$ (min) | $C_{m,a}$ (pcu/h) | $l_a$ (m) |
|---|---|---|---|
| 2-3 | 7.8 | 1500 | 300 |
| 2-5 | 16.2 | 1500 | 200 |
| 3-4 | 10.2 | 1300 | 300 |
| 3-6 | 6.0 | 1500 | 200 |
| 4-7 | 18.0 | 1300 | 200 |
| 5-6 | 10.2 | 1500 | 300 |
| 6-7 | 13.8 | 1500 | 300 |
| 3−2 | 16.2 | 1300 | 300 |
| 4−3 | 18.0 | 1300 | 300 |
| 5−2 | 10.2 | 1500 | 200 |
| 6−3 | 12.0 | 1500 | 200 |
| 6−5 | 13.8 | 1500 | 300 |
| 7−4 | 18.0 | 1300 | 200 |
| 7−6 | 13.8 | 1500 | 300 |

**Table 6. Travel information of entering/exiting connections.**

| Station connection arc | $l_a$ (m) | $T_{m1,a}$ (min) | $T_{m2,a}$ (min) |
|---|---|---|---|
| O-14 | 750 | 9.0 | 7.8 |
| O-8 | 600 | 7.2 | 7.2 |
| O-2 | 750 | 9.0 | – |
| 19-D | 750 | 9.0 | 7.8 |
| 13-D | 600 | 7.2 | 7.2 |
| 7-D | 750 | 9.0 | – |

**Table 7. Traffic assignment result of effective hyper-paths.**

| Optimal path | Path flow (pcu/h) | Travel impedance (min) |
|---|---|---|
| O-(O,14)-(14,15)-(15,18)-(18,19)-(19,D)-D | 963.5 | 17.4 |
| O-(O,14)-(14,15)-(15,16)-(16,19)-(19,D)-D | 938.6 | 18.6 |
| O-(O,14)-(14,17)-(17,18)-(18,19)-(19,D)-D | 924.8 | 19.2 |
| O-(O,8)-(8,9)-(9,10)-(10,13)-(13,D)-D | 924.3 | 15.0 |
| O-(O,8)-(8,9)-(9,12)-(12,13)-(13,D)-D | 890.9 | 15.6 |
| O-(O,8)-(8,11)-(11,12)-(12,13)-(13,D)-D | 816.4 | 16.8 |
| O-(O,2)-(2,5)-(5,6)-(6,7)-(7,D)-D | 814.0 | 18.0 |
| O-(O,2)-(2,3)-(3,6)-(6,7)-(7,D)-D | 797.8 | 17.4 |
| O-(O,2)-(2,3)-(3,4)-(4,7)-(7,D)-D | 679.6 | 16.2 |

The data in Table 7 illustrates that there is a indistinct difference in flow between disparate road sections. Compared to a distance path, the section's length that are shorter to the longest are apt to receive less traffic flow:

As depicted in Fig 7, the visualization of the iteration process is a scientific way to verify the effectiveness of the distribution algorithm: the relative gap of the objective output gradually approaches 0. Through the average of a series of auxiliary points in the iteration, the final iteration step is obtained after solving the auxiliary planning problem for a certain period of time, and the output is constantly corrected until it approaches the correct result. The iteration of traffic volume algorithm is to narrow the gap as well as quicken the calculation. The final iteration curve stabilizes at $5 \times 10^{-2}$ of gap value, and the total number of iterations has reached so far. In programming algorithms, the generation of line charts could visually present the optimization process and convergence tendency of the algorithm in each iteration, especially in the process of non-dominant solution set hierarchy, there exists a step of updating the dominance degree of individuals, thereby waiting for the algorithmic machine to select all levels of Pareto frontier, and laying the foundation for a new population's production in the next step.

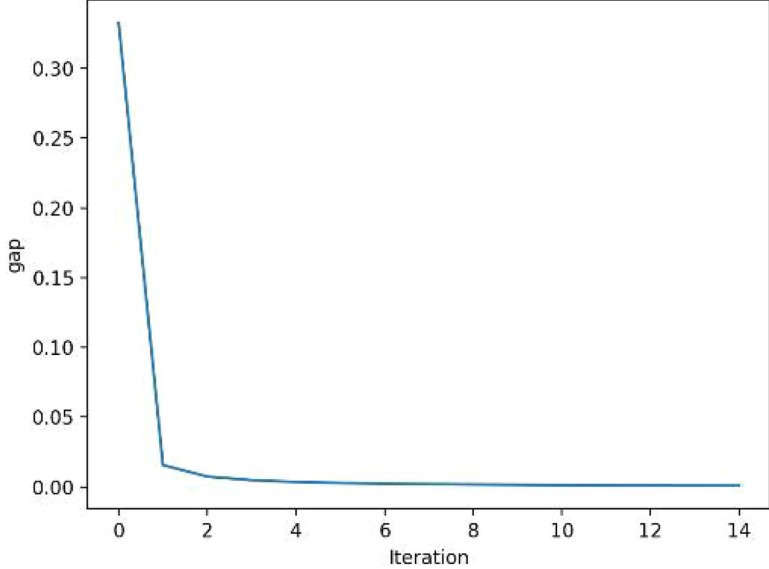

**Fig 7. Multi-modal network's path-searching iteration.**

**5.3.2. Pareto fronts' distribution.** Fig 8 illustrates that some variables, such as $v_m$, $k_m$, $A_{m,l'}$, $B_{m,l'}$, et al, which are deemed as irrelevant variables, they have nothing to do with the final results' distribution. The theoretical Pareto fronts, which contain both carrying capacity and individual travel time, are ought to form a smooth curve surface in three-dimensional space (Target 1, 2 are deemed to x and y axes respectively). As depicted in the figure below, the black-color curve represents the fitting curve of the optimal scattered point solution set, the fitting function is as follows (one decimal place retained after radix point): $y = 2.3 \times 10^{-12}x^3 + 1.9 \times 10^{-7}x^2 + 4.8 \times 10^{-3}x + 4.7 \times 10$; and the correlation coefficient of the objective solution is $9.8 \times 10^{-1}$. In summary, as individual users' travelling endurance decline, the plural modulus of the first objective is also dropping.

**5.3.3. Linear fitting.** In this section, the optimal solution group is selected to fit the bi-objective function. Fig 9 depicts the compromise fitting of the optimal result.

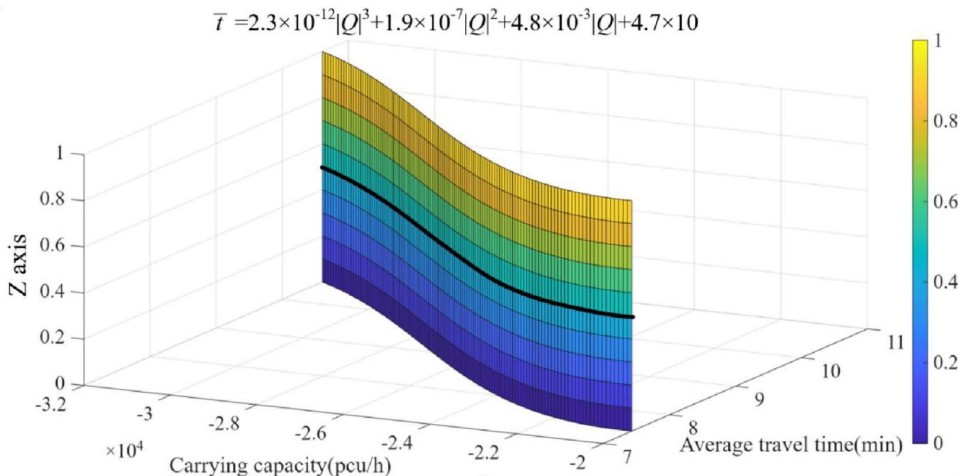

**Fig 8. Curved surface of multi-objective functions' optimization.**

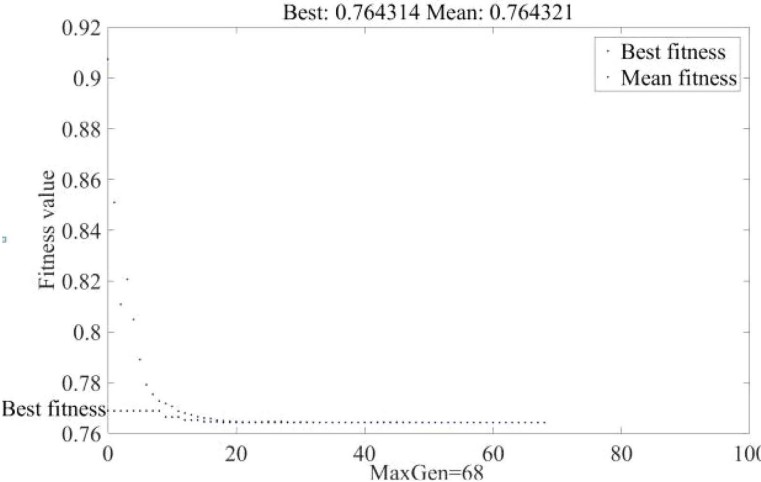

**Fig 9. The compromise solution fitting of optimal Pareto fronts.**

When the number of iterations reaches 68, the calculation results tend to be stable. The best fitness is between $7.6 \times 10^{-1}$ and $7.8 \times 10^{-1}$. Attain the matlab data for the genetic algorithm, and combine it with the expression of the bi-objective function. Weight coefficient calculation results show that when $\omega_1$ approaches 0.2 and there is a slight surplus of 0.8 for weight coefficient $\omega_2$, the optimal value of the compromise solution could be attained.

## 5.4. Sensitivity analysis

As shown in Fig 10. Apart from non-expandable black-marked sections, the rest red-marked 10 sections are all availably flexible to expand. The scheme includes station 15 to station 14; station 15 to station 18; station 16 to station 15; and station 8 to station 9,11; station 10 to station 9. In addition, the abstracted bike-sharing network covers 4 expandable road sections: station 4 to station 3,7; station 6 to station 3; station 3 to station 4.

It is worth mentioning that the calculation results of the improvement scheme in this paper are based on the above section-updated super network. The specific measure is to expand the capacity of the road mentioned above. Changes in the target results of this scheme before and after widening are partly shown in Table 8 [35], the essay adopted 25% value of the benchmark experimental threshold as an increment to launch the sensitivity analysis of the model. From these figures, there is a positive increment between bi-level planning model and the benchmark experiment.

As stated in Table 8, this scheme witnessed the growth of lane capacity. The indicator $a$ has gained its increment, whose value was denoted as 174, 69, and 81. Objectively, the change in carrying capacity has reached 2276.8pcu/h. Ultimately, the expansion cost consumed was fixed at 13505.74, with a unit of ten thousand yuan.

Besides, the triple-level planning model obtained a better solution on the basis of the previous promotion effect. Thus, a sensitivity analysis of NSGA-II's applicability was to provide an idea for the lanes' width broadening conditions. The expanded increment α in Table 8 was explicated as a reference, with broadening limit indicators set to 200, 250, 300, 350, and 400 pcu/h.

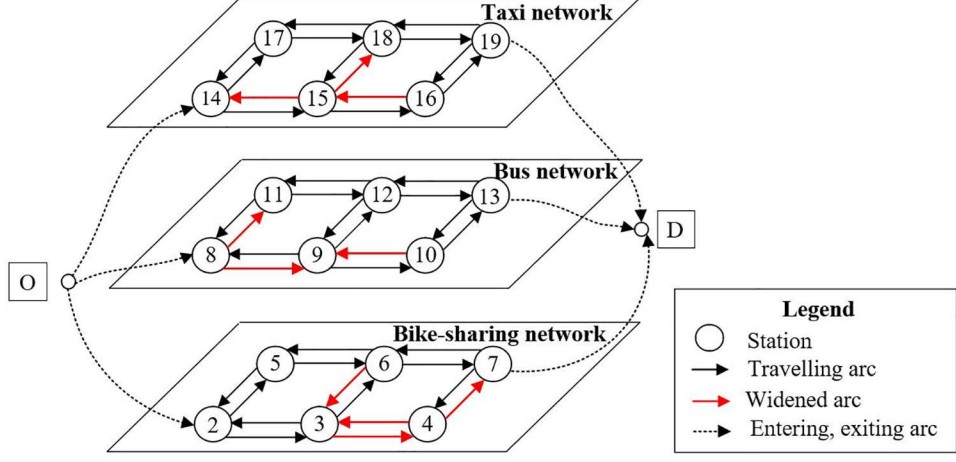

**Fig 10. The optimal widen scheme included in bi-directional road section's network.**

**Table 8. Benchmark experimental variations after expanding certain road sections.**

| Changes of variables | ΔQ (pcu/h) | Δα (pcu/h) | | | $\Delta g_a$ (10K) |
|---|---|---|---|---|---|
| | | Taxi | | Bus | Bike-sharing |
| Broaden limit is 200 | 2276.8 | 174 | 69 | 81 | 13505.7 |

**5.4.1. Upper threshold.** As shown in Fig 11, the promotion of widening threshold every 50 units mildly elevates the carrying capacity's modulus within the dependent variable ranging from 7.0 to 12.0. The fitting curve of Pareto fronts and their 95% confidence interval belt of the quartic polynomial function. For example, when the dependent variable along y axis equals 9.0, the corresponding carrying capacity's modulus indicates 3967.2pcu/h, 3761.3pcu/h, 3580.5pcu/h and 3345.8pcu/h separately from left to right, and the annotated figure presents the overall result (with dependent variable outweighing 7.0): $|Q_{kb=400}| > |Q_{kb=350}| > |Q_{kb=300}| > |Q_{kb=250}|$.

**5.4.2. Mobility rate.** With the increasing occupation of non-motor vehicle lanes by motor vehicle lanes on roads, aiming at simulating and distributing vehicles in the super network, this paper further compares while under three different levels of mobility rate: 0.7-0.8, 0.8-0.9 and 0.9-1.0, the contribution value of multi-objective function to carrying capacity. In Fig 12, $k_{motorized}$ is equal to the sum of sharing rates of cars and buses – $k_{taxi}$ ($k_1$) plus $k_{bus}$($k_3$). Obviously, the contribution

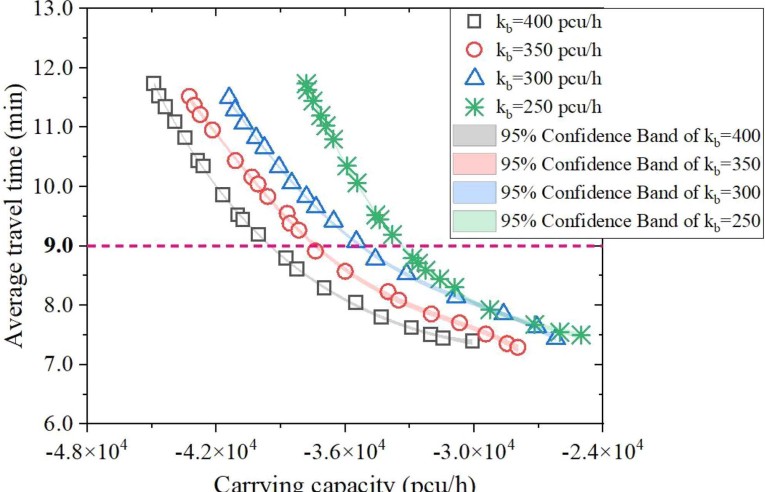

**Fig 11. Pareto frontier for every 50 units change in the upper threshold.**

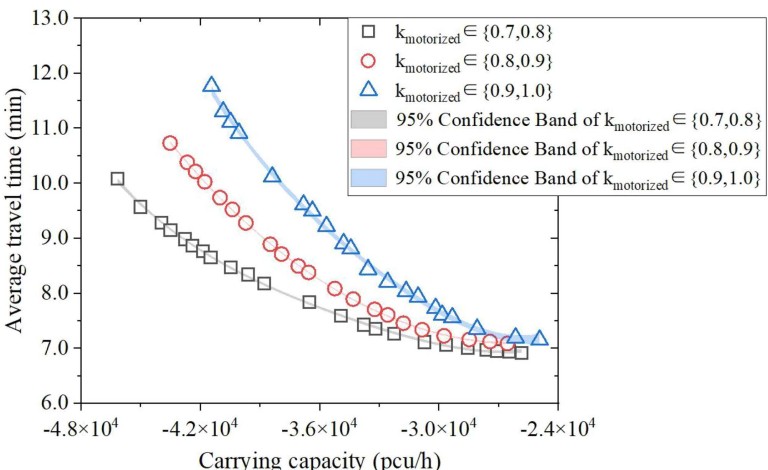

**Fig 12. Pareto frontier for every 0.1 units change in the motorized ratio.**

value of objective function to carrying capacity in three situations shows the same fluctuated law, which is mainly due to the fact that the number of motor vehicles is only related to the network topology and traffic model split, thus resulting in the fixed effects affecting the prediction way. In a word, for different motor vehicle ratios, the modulus of carrying capacity contribution value gradually decreases with the reduction of travel time. The three fitting curves ultimately stabilizes at $2.6 \times 10^4$ pcu/h, $2.7 \times 10^4$ pcu/h and $2.5 \times 10^4$ pcu/h respectively.

### 5.5. Countermeasures to promote carrying capacity

**5.5.1. Downgrade the level of service.** As described earlier, no matter what the traffic state was like, there would always be a corresponding optimal solution. Moreover, the scattered results would move further away from the origin as the saturation promoted by 0.1, which means that the carrying capacity would present a downward trend. Through taking a portion of Pareto Frontier, the fitting curve of Pareto fronts and their 95% confidence interval belt under three LOS conditions could be clearly presented. As shown in Fig 13.

Results show that the downgrading of average saturation elevates the carrying capacity of the network to a certain extent. For example, when the dependent variable along y axis equals 8.5 min, the corresponding carrying capacity modulus indicates 3813.7pcu/h, 3278.4pcu/h, 2779.6pcu/h respectively from left to right. and the annotated diagram presents the overall result (with dependent variable outweighing 7.0): $|Q_{\sigma=0.7}| > |Q_{\sigma=0.8}| > |Q_{\sigma=0.9}|$. In order to let correlation coefficient modulus $R^2$ according to the optimal solution close to 1, while combining the inherent curvature of Pareto Frontier. After error weaken in Origin Apps' Simple Fit, we get a series of quartic polynomial functions, label them as scientific notation with one decimal place retained, then fill in Table 9.

**5.5.2. Widen critical road sections** The countermeasures to broaden lane width could be separated into three sub-networks' transformation scheme.

Fig 14 shows the projection of Pareto Friontiers mapped on the xoy plane and verifies the obtained Pareto Friontiers in Fig 11.

## 6. Conclusion

With the aim of exploring the preferences of travelers' behaviors in a multi-modal network. This essay proposed a bi-level planning model to improve the carrying capacity by surveying road attributes and time indicators. Utilizing the

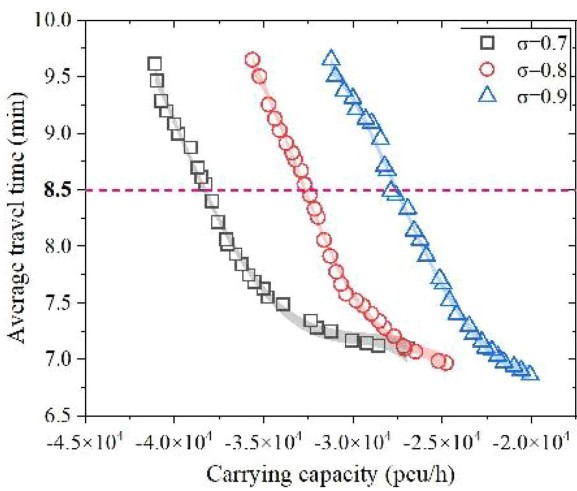

**Fig 13. Pareto fronts of stable LOS flow states.**

**Table 9. Objective function's average Pareto front of different LOS.**

| σ | Pareto fronts | Satisfied Pareto fronts | Changing trend of Polynomial fitting function |
|---|---|---|---|
| 0.7 | 28 | 28 | $y = -1.4 \times 10^{-16}x^4 - 2.0 \times 10^{-11}x^3 - 1.0 \times 10^{-6}x^2 - 2.3 \times 10^{-2}x + 1.8 \times 10^2$ |
| 0.8 | 29 | 28 | $y = -5.5 \times 10^{-16}x^4 - 6.6 \times 10^{-11}x^3 - 2.9 \times 10^{-6}x^2 - 5.6 \times 10^{-2}x - 4.0 \times 10^2$ |
| 0.9 | 30 | 30 | $y = -2.4 \times 10^{-16}x^4 - 2.3 \times 10^{-11}x^3 - 7.7 \times 10^{-7}x^2 - 1.1 \times 10^{-2}x - 5.5 \times 10^{-1}$ |

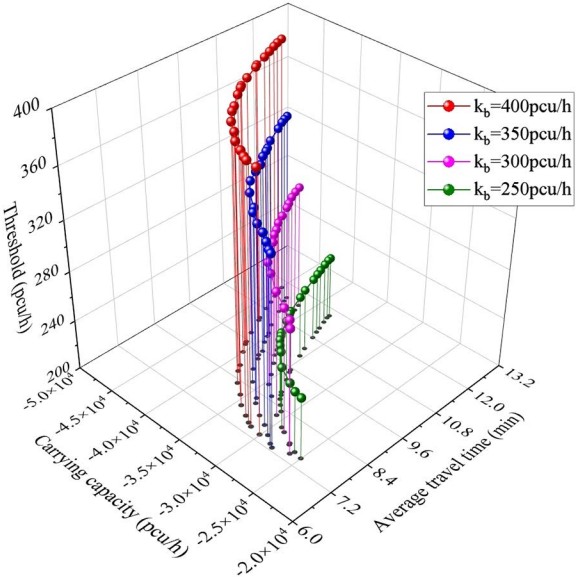

**Fig 14. Diagram of the fitted Pareto Friontiers' trajectory after expansion.**

non-dominated sorting genetic algorithm II to acquire the pareto fronts under different widening conditions, for this purpose, the flow distribution results achieved the optimal targeted weight coefficient by algorithm iteration. On the basis of that, the multi-objective and bi-level planning model was figured out through NSGA-II algorithm. Case study proved that traffic demand, average travel time, and whether the impedance function parameters change were all major factors affecting the carrying capacity's value. The plural modulus of carrying capacity in the super network grew accompanied with average travel time's rose. The sensitivity test verified its high-consistency transformation capability in comparison to the benchmark experiment.

However, there are still deficiencies in this eaasy, on account of quantitative factors are restricted by the accumulated process, such as economic development level and resident population, are difficult to be solely expressed by linear equations. Meanwhile, further study should explore and validate more parameters' definition in the transfer arcs of a directed network. In addition, index optimization and model accuracy are the key points to be improved in the future.

## Supporting information

**S1 File. S1_raw_images.**
(PDF)

**S1 Data. Supporting data.**
(ZIP)

## Author contributions

**Data curation:** Xiangyue Huang.

**Formal analysis:** Xiangyue Huang.

**Investigation:** Xiangyue Huang.

**Methodology:** Xiangyue Huang.

**Visualization:** Xiangyue Huang.

**Writing – original draft:** Xiangyue Huang.

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
