## [Decision Letter · Decision Letter 0]

24 Jul 2025

Dear Dr. Huang,

Thank you for submitting your manuscript to PLOS ONE. After careful consideration, we feel that it has merit but does not fully meet PLOS ONE’s publication criteria as it currently stands. Therefore, we invite you to submit a revised version of the manuscript that addresses the points raised during the review process.

We look forward to receiving your revised manuscript.

Kind regards,

Songpo Yang

Academic Editor

PLOS ONE

Journal Requirements: 

Reviewers' comments:

Reviewer's Responses to Questions

**Comments to the Author**

1. Is the manuscript technically sound, and do the data support the conclusions?

Reviewer #1: Yes

Reviewer #2: Yes

2. Has the statistical analysis been performed appropriately and rigorously?

Reviewer #1: Yes

Reviewer #2: Yes

3. Have the authors made all data underlying the findings in their manuscript fully available?

Reviewer #1: Yes

Reviewer #2: Yes

4. Is the manuscript presented in an intelligible fashion and written in standard English?

Reviewer #1: Yes

Reviewer #2: No

Reviewer #1: This paper is well organized and provides a scaling scheme for multi-modal networks, which lays the foundation for alleviating congestion at the macro level. Nevertheless, this paper still has some shortcomings the following issues need to be further described:

1.There are virtually no color images throughout the text, and tables are uniformly presented with traditional light gray borders rather than the three-line tables.

2.In “Data Collection” on pages 12 and 13, the font requirements for parameters and units are not uniform enough, and the font specification is somewhat arbitrary.

3.In the “Literature Review” on page 4, authors are advised to consider a separate narrative of the background of the study, including details and a summary of the established process of two-layer modeling.

4.In the Algorithm Design section on page 10, the key steps of the NSGA-II algorithm are not shown, i.e., the process of generating a new generation of populations from an elite strategy can be clearly illustrated with color pictures.

5.the authors state that "when the road saturation increases from 0.6 to 0.9, cyber carrying capacity increased by 13.97%, and average travel time of passengers reduced by 66.02%." What is the relationship between road saturation and road carrying capacity, and how does it significantly reduce passenger travel time? This is very confusing,please clarify.

6.In the author's abstract, 9 road segments are mentioned as research objects, but 10 road segments are mentioned in the conclusion, please explain.The author carefully checked and rewrote the conclusion section.

7.The author has mentioned the traffic carrying capacity many times in this paper, its unit is a negative number, and its order of magnitude and value are also various, so it is suggested to unify. Figure 12 What is the significance of the suggested use of a line chart?

Reviewer #2: 1. Lack of Clarity in Abstract, (e.g., "free-floating roads" on Page 1) and illogical "zero travel time" contradict real traffic conditions. Define key terms (e.g., "uncongested roads") and revise contradictions (e.g., "low-delay scenarios").

2. Assumption 3 (Page 11) ignores transfer time between modes, a critical factor in multimodal networks. Incorporate transfer time or justify its exclusion with references.

3. Data and Visualization Issues, Inconsistent units in Tables 3-6 (e.g., time in hours but values too small; 0.13h ≈ 7.8 min). Figure 7 (Page 20) fails to clearly illustrate correlations, standardize units (e.g., minutes) and enhance graphs with labels/trend analysis.

4. Add parameter sensitivity tests or cite benchmark studies; include comparative experiments.

5. Abstract claims "13.97% capacity increase when σ=0.6→0.9", but the conclusion (Page 25) states "|Q| at σ=0.9 is lower than at σ=0.7/0.8". verify data consistency and revise conclusions accordingly.

6. Incorrect tense in abstract ("this study solely surveyed" → "examined"). Author "Younghun B" in [9] should be "B, Y."; DOI in [13] contains extra spaces.

**Do you want your identity to be public for this peer review?** For information about this choice, including consent withdrawal, please see our Privacy Policy

Reviewer #1: No

Reviewer #2: No

---

## [Author Response · Author response to Decision Letter 1]

23 Aug 2025

I would like to receive any comments from the journal office. I have also addressed the specific concerns raised by each reviewer and made necessary adjustments to enhance the clarity and quality of our work. Additionally, I clarified some sections for better understanding and added relevant references where needed. I believe these revisions significantly improve my revised manuscript, making it more robust and comprehensive. Thank you once again for this opportunity; I look forward to your feedback on the revised submission.

---

## [Decision Letter · Decision Letter 1]

9 Sep 2025

Dear Dr. Huang,

Thank you for submitting your manuscript to PLOS ONE. After careful consideration, we feel that it has merit but does not fully meet PLOS ONE’s publication criteria as it currently stands. Therefore, we invite you to submit a revised version of the manuscript that addresses the points raised during the review process.

We look forward to receiving your revised manuscript.

Kind regards,

Songpo Yang

Academic Editor

PLOS ONE

Journal Requirements:

Reviewers' comments:

Reviewer's Responses to Questions

**Comments to the Author**

Reviewer #2: All comments have been addressed

2. Is the manuscript technically sound, and do the data support the conclusions?

Reviewer #2: Yes

3. Has the statistical analysis been performed appropriately and rigorously?

Reviewer #2: Yes

4. Have the authors made all data underlying the findings in their manuscript fully available?

Reviewer #2: No

5. Is the manuscript presented in an intelligible fashion and written in standard English?

Reviewer #2: Yes

Reviewer #2: The article points out that "when the road saturation increases from 0.7 to 0.9, the bearing capacity modulus decreases by 27.1%," but it does not clearly explain the implications of this change for actual traffic management policies. The author is requested to further elaborate: (1) Does this conclusion imply that traffic flow should be actively restricted during peak hours to maintain lower saturation? (2) Is the cost-benefit analysis of widening key road sections (such as the 10 sections mentioned in the article) sufficient to support their prioritization as improvement projects? Have other intervention measures (such as signal optimization and bus priority) been considered?

**Do you want your identity to be public for this peer review?** For information about this choice, including consent withdrawal, please see our Privacy Policy

Reviewer #2: No

---

## [Author Response · Author response to Decision Letter 2]

26 Oct 2025

Dear reviewers:

The author has thoughtfully addressed each of the reviewers' suggestions and expresses sincere gratitude to both reviewers for their invaluable feedback on the 2nd manuscript. The depth and rigor of the reviewers' analyses, as evidenced by the questions raised, highlight their significant practical relevance. After careful consideration, the responses to these points have been incorporated into the revised version based on the previous draft. The response letter accompanying the revision will include solutions directly within each textual explanation, accompanied by pertinent references. The 3rd revised manuscript will also continue to rectify any inappropriate phrasing identified in the article. Feedback from both reviewers regarding textual revisions is summarized as follows: The revised paper employs the present tense, past tense, and past perfect tense for narrative clarity. Primary, secondary, and tertiary headings are formatted in Times New Roman 18pt, 14pt, and 11pt, respectively, while figure and table captions consistently utilize Times New Roman 10pt.

---

## [Decision Letter · Decision Letter 2]

2 Dec 2025

Title-Modelling and Improving Approach for Carrying Capacity in Multi-modal Super Network by Considering Travel Time

PONE-D-25-28716R2

Dear Dr. Huang,

We’re pleased to inform you that your manuscript has been judged scientifically suitable for publication and will be formally accepted for publication once it meets all outstanding technical requirements.

Kind regards,

Songpo Yang

Academic Editor

PLOS ONE

Additional Editor Comments (optional):

Reviewers' comments:

Reviewer's Responses to Questions

**Comments to the Author**

Reviewer #2: (No Response)

2. Is the manuscript technically sound, and do the data support the conclusions?

Reviewer #2: (No Response)

3. Has the statistical analysis been performed appropriately and rigorously?

Reviewer #2: (No Response)

4. Have the authors made all data underlying the findings in their manuscript fully available?

Reviewer #2: (No Response)

5. Is the manuscript presented in an intelligible fashion and written in standard English?

Reviewer #2: (No Response)

Reviewer #2: (No Response)

**Do you want your identity to be public for this peer review?** For information about this choice, including consent withdrawal, please see our Privacy Policy

Reviewer #2: No

---

## [Editor Report · Acceptance letter]

PONE-D-25-28716R2

PLOS One

Dear Dr. Huang,

I'm pleased to inform you that your manuscript has been deemed suitable for publication in PLOS One. Congratulations! Your manuscript is now being handed over to our production team.

Kind regards,

on behalf of

Dr. Songpo Yang

Academic Editor

PLOS One